# A High-Speed Demodulation Technology of Fiber Optic Extrinsic Fabry-Perot Interferometric Sensor Based on Coarse Spectrum

**DOI:** 10.3390/s21196609

**Published:** 2021-10-03

**Authors:** Peng Zhang, Ying Wang, Yuru Chen, Xiaohua Lei, Yi Qi, Jianghua Feng, Xianming Liu

**Affiliations:** Key Lab of Optoelectronic Technology and Systems Ministry of Education, College of Optoelectronic Engineering, Chongqing University, Chongqing 400044, China; WangY2019@cqu.edu.cn (Y.W.); chenyuru@cqu.edu.cn (Y.C.); xhlei@cqu.edu.cn (X.L.); qiyi2014@126.com (Y.Q.); fengjianghuaasd@163.com (J.F.); xianming65@163.com (X.L.)

**Keywords:** extrinsic Fabry-Perot interferometers, high-speed demodulation, coarse spectrum, maximum likelihood estimation

## Abstract

A fast real-time demodulation method based on the coarsely sampled spectrum is proposed for transient signals of fiber optic extrinsic Fabry-Perot interferometers (EFPI) sensors. The feasibility of phase demodulation using a coarse spectrum is theoretically analyzed. Based on the coarse spectrum, fast Fourier transform (FFT) algorithm is used to roughly estimate the cavity length. According to the rough estimation, the maximum likelihood estimation (MLE) algorithm is applied to calculate the cavity length accurately. The dense wavelength division multiplexer (DWDM) is used to split the broadband spectrum into the coarse spectrum, and the high-speed synchronous ADC collects the spectrum. The experimental results show that the system can achieve a real-time dynamic demodulation speed of 50 kHz, a static measurement root mean square error (RMSE) of 0.184 nm, and a maximum absolute and relative error distribution of 15 nm and 0.005% of the measurement cavity length compared with optical spectrum analyzers (OSA).

## 1. Introduction

Extrinsic Fabry-Perot interferometers (EFPI) sensor has the advantages of strong immunity to electromagnetic interference, high sensitivity, compact structure [1,2], good stability, high resolution [3], thermally insensitive [4], low-cost equipment [5] and resistance to the harsh environment [6]. Comparing to the intrinsic Fabry-Perot interferometer (IFPI), the cavity of EFPI can be opened to the environment [7]. It is used to measure the transient signal [8], such as the shock wave overpressure in the exploding field [9].

At present, the intensity demodulation method [10,11,12,13,14,15,16] is a common high-speed measurement method for EFPI sensors. The relationship between the intensity of a single or multiple wavelengths and cavity length of the EFPI is used to realize signal demodulation. For example, the three-wavelength intensity demodulation method proposed by Markus Schmidt has a speed of 80 kHz [17,18], which meets the requirements of transient testing. However, the measurement range is limited, and precision is susceptible to noise. Compared with the intensity demodulation method, spectral demodulation method uses the relationship between the frequency of the interference fringe and cavity length of the EFPI. Because the frequency spectrum is independent of the source intensity fluctuations, this method can effectively suppress the influence of light source intensity fluctuation and transmission loss. In order to reduce the impact of noise and improve the demodulation accuracy, a variety of effective methods were proposed, such as frequency estimation method [19], wavelet phase extraction algorithm [6], instantaneous phase extraction method [20], wavelet transform and polarization low coherence interferometry [21]. However, all methods above rely on a densely sampled spectrum, which mainly affects the demodulation speed.

Therefore, this paper proposes a method of reducing the spectral acquisition density to improve the demodulation speed. A dense wavelength division multiplexer (DWDM) is used to split the EFPI spectral into the multiple optical paths in the light frequency domain. An analog-to-digital converter (ADC) is used to convert optical signals into digital signals in each optical path synchronously. What is more, the joint algorithm of fast Fourier transform (FFT) and maximum likelihood estimation (MLE) can guarantee demodulation accuracy at high demodulation speed. The demodulation principle of the coarse spectrum is analyzed and verified by experiments.

## 2. Theory

### 2.1. Coarse Spectral Sampling Point Analysis

The traditional EFPI sensor demodulation system [22] is shown in Figure 1. According to the principle of two-beam interference, the light intensity *I_r_*(*λ*) of the reflected light of the EFPI sensor is:(1)$$I_{r}(\lambda )=2R[1-\cos(\frac{4\pi L}{\lambda })]I_{0}(\lambda )$$
where *R* is the reflectivity of both ends of the Fabry-Perot cavity, *I*_0_(*λ*) is the intensity of incident light, *L* is the length of the Fabry-Perot cavity and *λ* is the wavelength of light.

After filtering out the direct current (DC) element, *I_r_*(*λ*) can be written as Equation (2):(2)$$I_{n}=A\cos(2\pi \cdot \frac{2L}{c}\cdot \nu_{n})+\omega_{n}$$
where *I_n_* refers to samples in the spectral data *I_r_*(*λ*), *c* is the speed of light, *υ_n_* is the frequency of light and *λ* = *υ/c*. *A* is the intensity coefficient and *ω_n_* is the Gaussian white noise.

Since *υ_n_* is an independent variable in Equation (2), the interference fringe frequency *f* = 2*L/c* can be obtained by comparing Equation (2) with cos(*2πfυ_n_*). It shows that the demodulation of cavity length is essentially an estimation of the frequency of the cosine signal in Equation (2).

In order to accurately obtain the frequency of the interference fringe of the spectrum, according to the Nyquist sampling theorem, the sampling frequency *f_s_* must be greater than 2 times the maximum signal frequency *f*. In addition, the sampling frequency *f_s_* is equal to (*Nλ*_min_*λ*_max_)/(*Bc*), where *B* is the spectral bandwidth, *N* is the sampling number, *λ*_min_, *λ*_max_ is the minimum and the maximum wavelength of the source. The sampling points *N* can be calculated as Equation (3):(3)$$N\geq \frac{4BL}{\lambda_{\min}\lambda_{\max}}$$

According to the parameters of the C-band source (the spectral bandwidth *B* is 40 nm (*λ*_min_ = 1525 nm, *λ*_max_ = 1565 nm)) and the initial cavity length of the EFPI sensor of 500 μm, the number of sampling points is *N* ≥ 33.5. Therefore, a 40-channels DWDM with a frequency interval of 12.75 GHz is used to split the broadband spectrum into the coarse spectrum.

### 2.2. The Joint Algorithm of FFT and MLE

In order to analyze the relationship between spectral sampling points and demodulation accuracy, the demodulation simulation is carried out for 2020 points and 40 points spectrum. The dense spectrum with 2020 points is shown as the blue line in Figure 2a. The coarse spectrum obtained by the 40-channels DWDM is shown as the red dots in Figure 2a. As shown in Figure 2b, compared with the dense spectrum, the intensity and resolution of the coarse spectrum decrease, but the frequency corresponding to the peak magnitude in Figure 2b does not change greatly.

Suppose *f_c_* is the frequency corresponding to the maximum magnitude of the FFT of the coarse spectrum. The rough estimation of the cavity length can be obtained by:(4)$$L_{\mathrm{FFT}}=\frac{c\cdot f_{c}}{2}$$

The purpose of reducing the amount of data is to reduce the spectral sampling time, but with the reduction of spectral sampling points, the resolution of the FFT decreases seriously. Therefore, on the basis of FFT demodulation results, the MLE algorithm [23] is introduced to improve the demodulation accuracy. In addition, when the search range of cavity length is small, the computed amount of the MLE algorithm will be reduced so that a faster demodulation speed can be obtained.

The MLE function can be written as:(5)$$\mathrm{MLE}\left(I_{n},L^{\prime }\right)=\max\left\{\sum_{n=0}^{N-1}I_{n}\cos\left(4\pi \frac{\nu_{n}}{c}L^{\prime }\right)\right\}$$
where *L*′ is the length of a constructed cavity. The cosine function in Equation (5) can be expressed as the real part of a complex exponential function. The light frequency can be written as *υ_n_* = *υ*_0_ + *nδυ*, where *υ*_0_ is the initial light frequency of the coarse spectrum and *δυ* is the interval of light frequency. The MLE function can be rewritten as:(6)MLE(In,L′)=∑n=0N−1InRe[e−j4πL′c(ν0+nδν)]=Re[e−j4πL′cν0∑n=0N−1Ine−j4πL′cnδν]=Re[e−j4πL′cν0F(In)]
where *n* is the number of samples, *n* = 1, 2, …, *N*. According to Equation (6). The FFT function and MLE function of the EFPI coarse spectrum is shown in Figure 3a.

The detailed diagram corresponding to the spectral peak is shown in Figure 3b. Compared with the FFT function, the MLE function can guarantee the resolution and demodulation accuracy under the condition of coarse spectral sampling. The shortcoming lies in the rapid increase of computation, which affects the real-time demodulation speed of the system. In addition, the relationship between the demodulation time of MLE algorithm and the peak seeking interval and peak seeking range of MLE is simulated by a laboratory computer (Processor: Intel (R) Core (TM) i5-8400; CPU: 2.8 GHz; RAM: 8 GB). The simulation results are shown in the following Table 1.

In order to reduce the amount of computation, the proposed algorithm includes two steps: rough estimation and fine estimation, as shown in Figure 4. Firstly, the intensity spectrum corresponding to the cavity length of EFPI is obtained by the fast FFT. Then, the rough cavity length *L_CS_* corresponding to the spectral peak is obtained by cubic spline interpolation, shown in Figure 4a. The MLE function is calculated in a narrow range *R_MLE_* based on *L_CS_* shown in Figure 4b. Finally, after spline interpolation, the precise estimation *L_MLE_* of cavity length is obtained.

### 2.3. Signal-to-Noise Ratio Analysis

With the reduction of spectral resolution and sampling points, the FFT fence effect becomes more significant, and noise will affect the determination of peak frequency. When the signal-to-noise ratio (SNR) of the spectral signal is lower than a certain threshold, the demodulation result will have a jump error [24]. Therefore, it is necessary to analyze the SNR requirement of spectral signals.

Assume that the noise distribution follows the Gaussian white noise, where *ω_i_* ~ *N* (0, *σ*^2^). The expression of known SNR is:(7)$$\mathrm{SNR}=10\mathrm{lg}\frac{A^{2}}{2\sigma^{2}}$$

Cramer–Rao lower bound (CRLB) [25,26,27,28] is the theoretical minimum error of signal parameter estimation with certain noise. Therefore, the noise limit of the algorithm can be represented by the CRLB bound, so as to determine the SNR requirement of the system. The relationship between CRLB of FFT and MLE fusion algorithm and SNR:(8)$${\mathrm{CRLB}}_{\mathrm{MLE}}=\frac{\lambda_{c}{}^{2}}{N}\cdot \frac{1}{\mathrm{SNR}}$$

In Equation (8), *λ_c_* represents the center wavelength, and *N* represents the number of sampling points. Monte Carlo simulation experiment [29] is used to simulate the demodulation Error of the fusion algorithm, which is represented by root mean square error (RMSE). The relevant parameters such as the central wave number, spectral bandwidth and sampling point (40 points) of the C-band light source are substituted. The simulation results of demodulation error are shown in Figure 5.

Figure 5 shows that when the demodulation error reaches the CRLB of the fusion algorithm, the corresponding SNR is about 20 dB. Therefore, The SNR of the demodulation system should be greater than 20 dB.

### 2.4. Measurement Ranges Analysis

At the same time, the measurement range will be affected when the samples drop. Next, the upper and lower limits of the measurement range are analyzed, respectively.

Since the Fourier transform peak is the envelope of the MLE peak, the spectral leakage error is only related to the spectral information of the signal. Therefore, the Fourier transform algorithm does not affect the analysis results. Due to the limitation of the bandwidth of the light source, the interference spectrum is a truncated signal. The influence of truncated signals on signal demodulation lies in spectrum leakage. The degree of leakage is not only related to the spectral amplitude but also related to the length of the EFPI cavities to be demodulated. In a certain bandwidth range, the larger the length of the EFPI cavity, the more interference fringes in the spectrum, and the smaller the influence of spectrum leakage. When the length of the EFPI cavity is small to a certain extent, the influence of spectrum leakage will be greater than the influence of random noise. At this point, it can be considered that it is impossible to achieve accurate demodulation of the interference spectrum. Therefore, the criterion for solving the EFPI cavity length minima is that the influence of spectrum leakage is less than that of random noise. Specifically, it is reflected as the sidelobe between the two main peaks in frequency close to one of the main peaks, whose amplitude is less than the amplitude of noise, as shown in Figure 6.

According to the criterion for solving the minimum EFPI cavity length and the error 3*σ* criterion, the expression of the minimum EFPI cavity length in the wavelength domain can be obtained:(9)$$L_{\min}=\frac{\left\{2\left\lceil \frac{\sqrt{2\mathrm{SNR}}}{6\pi }-\frac{1}{2}\right\rceil +1\right\}\lambda_{c}{}^{2}}{8B}$$

The minimum EFPI cavity length is related to SNR and spectral bandwidth *B*. The minimum EFPI cavity length increases with the increase of SNR or the decrease of spectral bandwidth. When the C-band light source is used, the wavelength is uniformly sampled at 40 points. The simulation results of the relationship between the minimum EFPI cavity length and SNR are shown in Figure 7.

As can be seen from Figure 7, the minimum EFPI cavity length remains unchanged within a certain SNR range. When the C-band broadband light source is adopted and the SNR is 20 dB, the lower limit of the theoretical measurement range of the system is *L*_min_ = 25.66 μm. The relationship between the maximum EFPI cavity length, sampling number and spectral bandwidth can be derived from Equation (3):(10)$$L_{\max}=\frac{N\cdot \lambda_{c}{}^{2}}{4B}$$

That is, the maximum demodulation EFPI cavity length is related to the spectral bandwidth *B* and sampling points *N*. The maximum EFPI cavity length increases with the increase of the central wavelength or the decrease of the spectral bandwidth. The simulation results of the relationship between the maximum demodulated EFPI cavity length and sampling points are shown in Figure 8.

As can be seen from Figure 8, within a certain spectral bandwidth range, the maximum demodulation EFPI cavity length will increase with the increase of sampling points. Plus, it is basically linear. Under the same conditions, the upper limit of the theoretical measurement range of the system is *L*_max_ = 684.13 μm.

Finally, the theoretical measurement range of the system is:(11)25.66 μm≤L≤684.13 μm

## 3. Experiments and Results

### 3.1. Experimental Measurement System

The experimental measurement system is shown in Figure 9. Experimental system one is a demodulation system for testing, and experimental system two is an ASE light source and an optical spectrum analyzer (OSA) system. The two systems are switched by an optical switch. The EFPI sensor used in the experiment is composed of a single-mode optical fiber and a mirror bonded to the PZT (piezoelectric transducer). The end of the SMF is polished and fixed on the displacement platform. The cavity length of the EFPI can be adjusted continuously. The mirror is attached to a piezoelectric transducer (PZT) and modulated by a driver signal. The output voltage of the precision voltage source is used to drive the PZT to change the length of the EFPI sensor cavity. The maximum shape variable of the PZT is 7 μm, and the maximum driving voltage is 150 V.

The components of the proposed demodulation system (system one) are shown in Figure 9c. The ASE broadband light is transmitted through the circulator to the EFPI sensor. The reflected signal of the EFPI sensor is split into coarse spectrum of 40 points by the 40-channels DWDM. Finally, the high-speed synchronous ADC and photoelectric array are used to sample the electrical signals and transmit them to the computer (Processor: Intel I CoITM) i5-8400, CPU:2.8 GHz, RAM: 8 GB)) through the gigabit network. The computer processes the uploaded data in real-time and calculates the cavity length of the EFPI sensor through the joint algorithm of MLE and FFT. After testing, the SNR of the demodulation system is about 26.75 dB, which meets the requirement of 20 dB SNR.

The OSA system (system two) includes an ASE broadband light and a spectrometer (The bandwidth is 40 nm and the resolution is 20 pm).

### 3.2. Static Measurement Experiment

A static continuous cavity length measuring experiment is conducted. The measurement object is an EFPI sensor with a fixed cavity length. The proposed demodulation system (system one) and the OSA system are connected to the EFPI sensor by an optical switch. Then, the demodulation system demodulates 50,000 cavity lengths in real-time, and the spectrometer system demodulates 1000 cavity lengths with the same algorithm.

Figure 10a shows the real-time demodulation results of 50,000 spectrums of the demodulation system, with an average value of 266.38946 μm and a root mean square error (RMSE) of 0.184 nm. Figure 10b shows the demodulation results of 1000 groups obtained by the spectrometer using the same algorithm, with an average value of 266.39015 μm and an RMSE of 0.143 nm. The difference between the measurement results of the two systems on the same sensor is only 0.69 nm. It can be considered that the coarse spectral demodulation system proposed in this paper, and the dense sampling system have basically the same measurement accuracy.

### 3.3. Dynamic Real-Time Measurement Experiment

The dynamic real-time measurement is conducted. The signal generator and the power amplifier generate a sinusoidal alternating current (AC) voltage signal of a given frequency. The cavity length of the EFPI is sinusoidal, corresponding to the voltage signal, and the real-time cavity length *L* is obtained through the demodulation system.

Figure 11 shows the real-time cavity length obtained by the demodulation system when the PZT is driven by a sinusoidal signal of 2 kHz. The real-time demodulation speed of the system is verified to be 50 kHz.

### 3.4. Comparison Experiment of Different System and Algorithm

In order to verify the accuracy of the demodulation system when the cavity length of EFPI sensor variation is small. The demodulation system is demodulated by FFT and the joint algorithm of the FFT and MLE, and the results are compared with the OSA system shown in Figure 12.

As can be seen from Figure 12, in the spectral coarse sampling system, the introduction of the MLE algorithm can greatly improve the demodulation accuracy and resolution of the FFT algorithm alone. At the same time, with the same FFT and MLE algorithm, the coarse spectral system and the dense spectral system have similar demodulation accuracy and resolution. Under the same algorithm, by comparing the demodulation results of the proposed system and OSA system, the maximum absolute error and relative error are 15 nm and 0.005%, respectively.

## 4. Conclusions

With the coarse spectral sampling technique, the speed of the demodulation system can be greatly improved while the demodulation accuracy does not decrease significantly. The experimental results show that the EFPI demodulation system based on coarse spectrum adopts joint algorithm, and the real-time measurement speed can reach 50 kHz, and the static measurement RMSE is 0.184 nm. The demodulation accuracy of a fusion algorithm is higher than that of the FFT algorithm alone. Compared with The OSA system, the relative error of cavity length measurement is less than 0.005%. In the case of low real-time demand, the spectral data can be stored first and then processed. This spectral data processing method can obtain a higher demodulation speed. The potential of the proposed demodulation method for transient testing is verified.

## Figures and Tables

**Figure 1 sensors-21-06609-f001:**
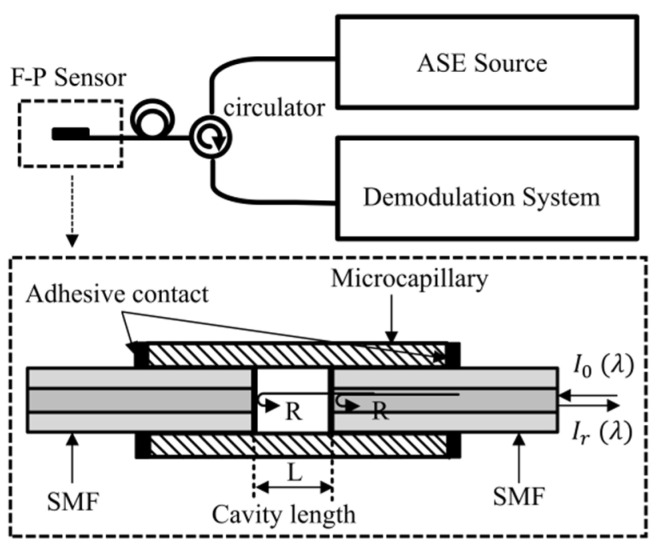
Fiber optic extrinsic Fabry-Perot interferometer demodulation system. (ASE: amplified Sspontaneous emission; F—P Sensor: Fabry-Perot sensor; SMF: single-mode fiber; R: optical fiber end reflectivity; *I*_0_(*λ*): incident light intensity; *I_r_*(*λ*): intensity of reflected light).

**Figure 2 sensors-21-06609-f002:**
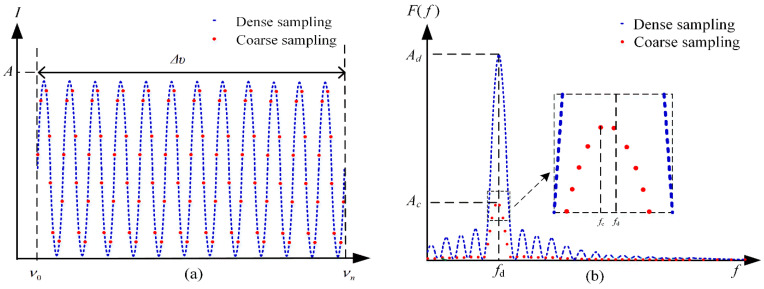
(**a**) The spectrum of dense sampling and coarse sampling and (**b**) the FFT results of the spectrum of dense sampling and coarse sampling.

**Figure 3 sensors-21-06609-f003:**
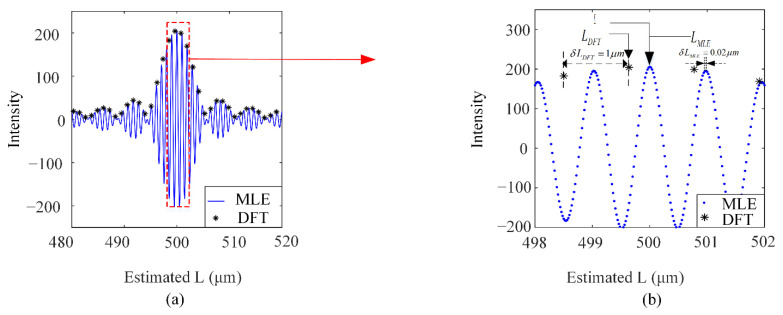
(**a**) The spectrum of FFT and MLE and (**b**) detail of the corresponding spectral peaks (Assuming *L* = 500 μm).

**Figure 4 sensors-21-06609-f004:**
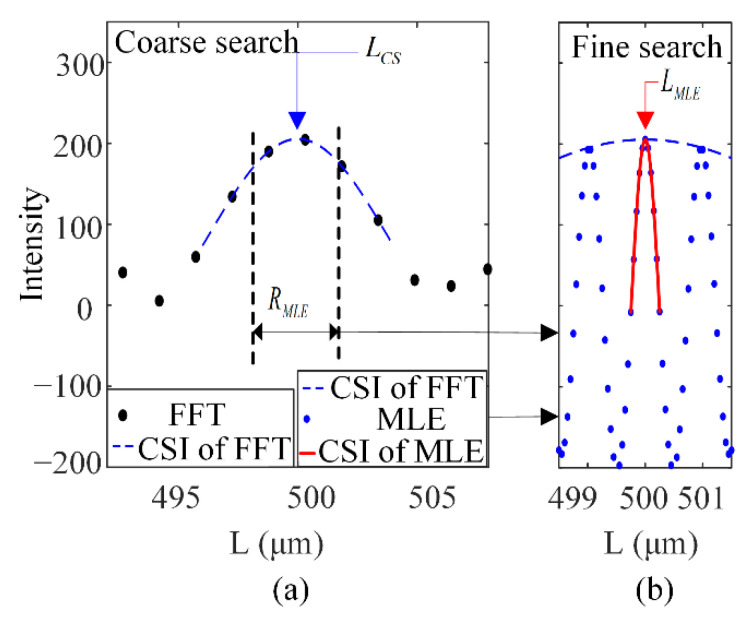
The implementation steps of the MLE algorithm: (**a**) FFT rough estimate and (**b**) MLE fine estimate (Assuming *L* = 500 μm).

**Figure 5 sensors-21-06609-f005:**
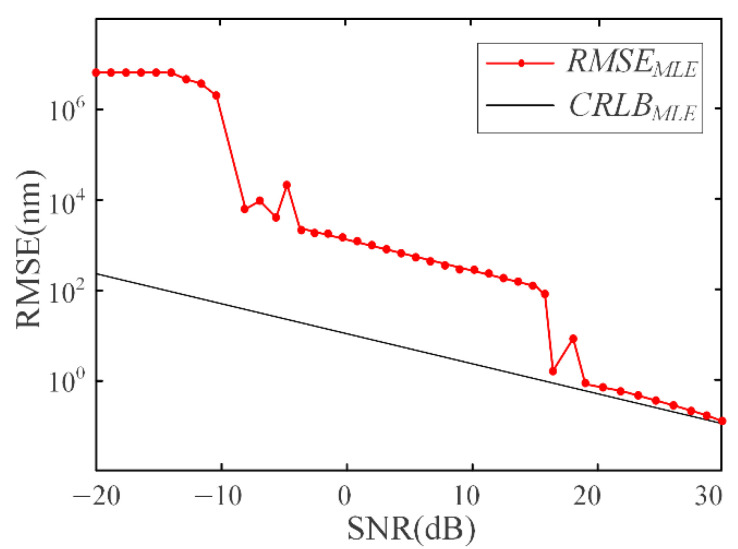
Root mean square error and Cramer–Rowe lower bound under different SNR.

**Figure 6 sensors-21-06609-f006:**
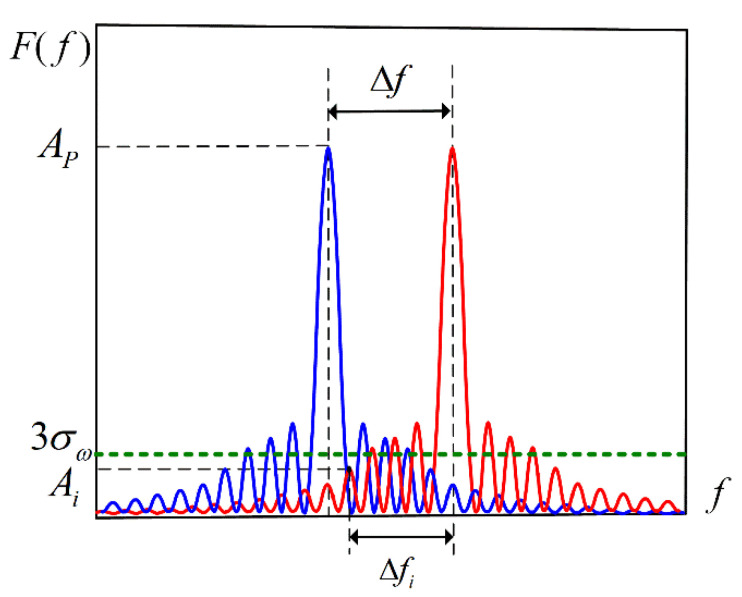
The schematic diagram of spectrum error analysis.

**Figure 7 sensors-21-06609-f007:**
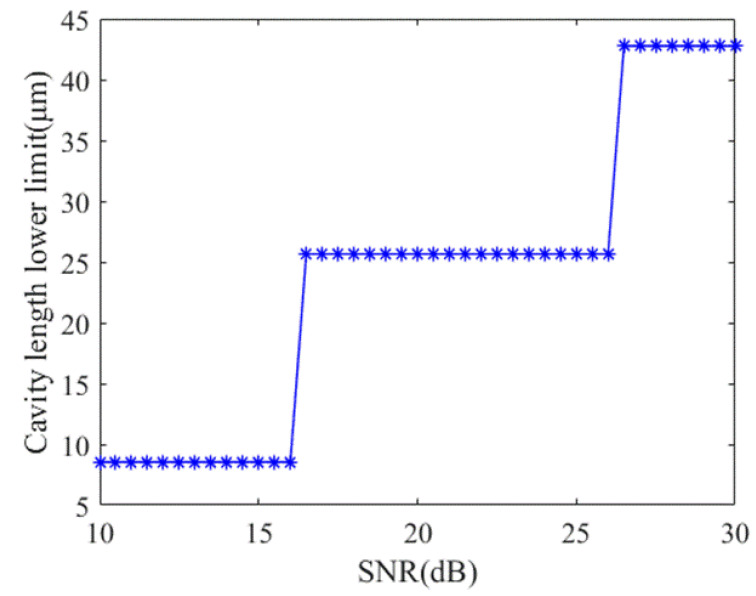
The relationship between minimum EFPI cavity length and SNR.

**Figure 8 sensors-21-06609-f008:**
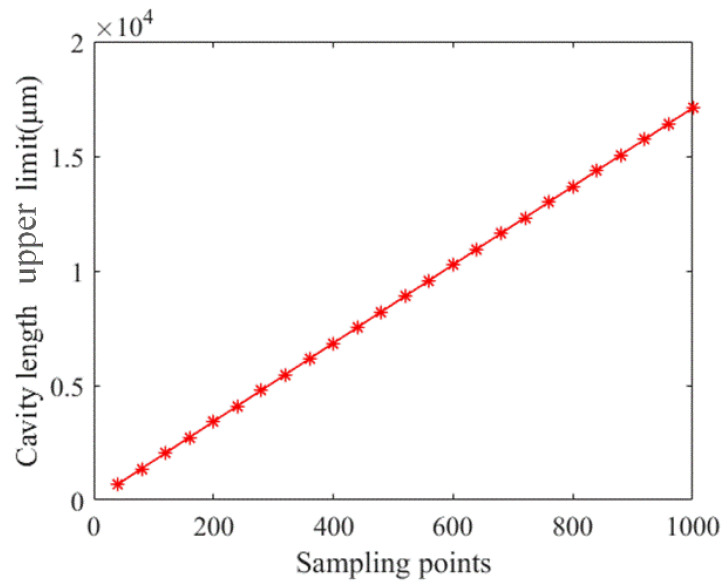
The relationship between the maximum EFPI cavity length and the number of sampling points.

**Figure 9 sensors-21-06609-f009:**
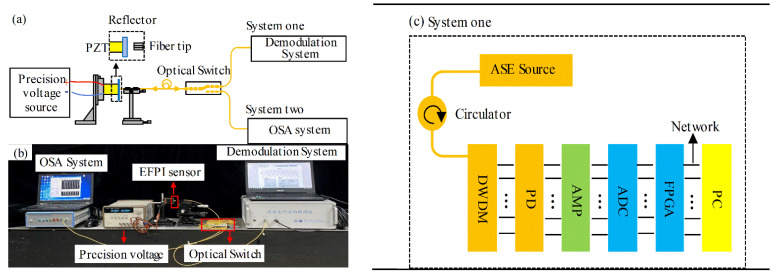
(**a**) Schematic of experimental measurement system, (**b**) photograph of the experimental measurement system and (**c**) schematic of system one (the proposed system) (PZT: piezoelectric transducer; ASE: amplified spontaneous emission; DWDM: dense wavelength division multiplexing; PD: photodetector; AMP: amplifier; ADC: analog-to-digital converter; FPGA: field programmable gate array (Xilinx Spartan-6, XC6SLX45-2FGG4841); PC: personal computer (processor: Intel (R) Core (TM) i5-8400; CPU: 2.8 GHz; RAM: 8 GB)).

**Figure 10 sensors-21-06609-f010:**
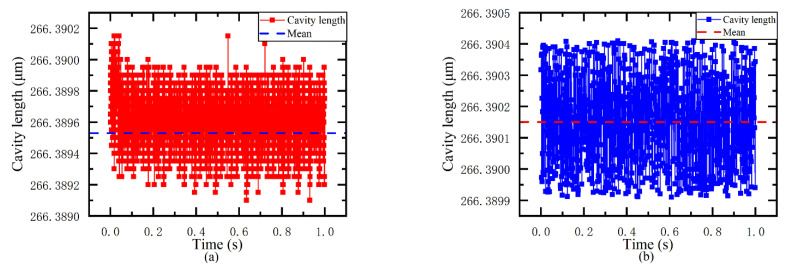
(**a**) Demodulation results of the demodulation system and (**b**) demodulation results of the spectrometer system.

**Figure 11 sensors-21-06609-f011:**
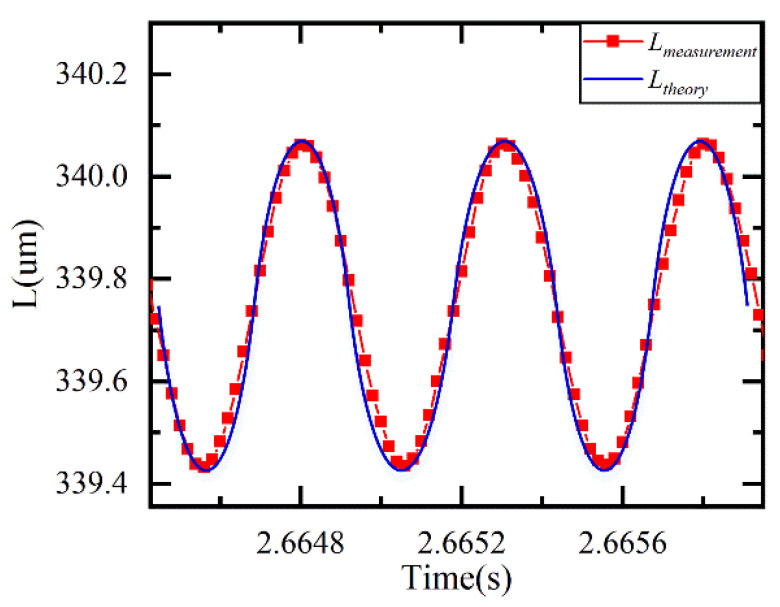
Dynamic real-time measurement results.

**Figure 12 sensors-21-06609-f012:**
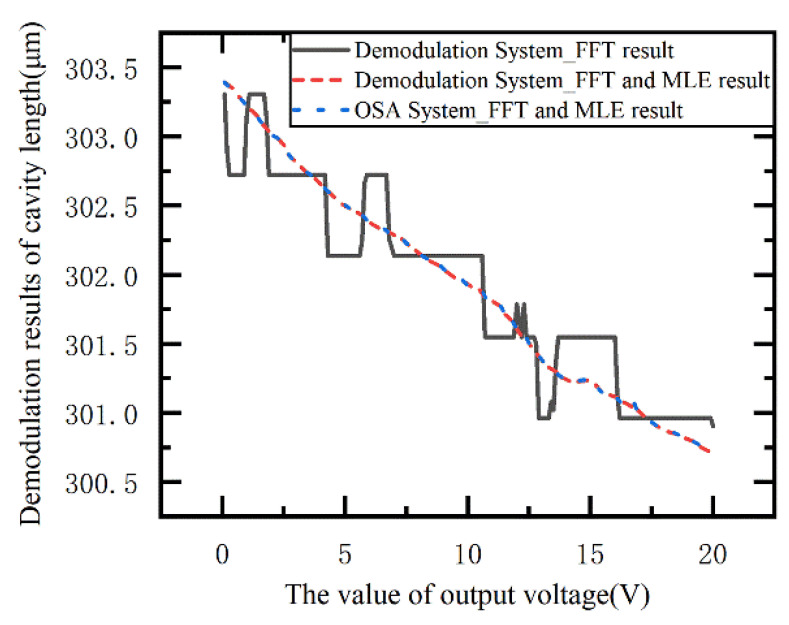
The demodulation results between OSA and the demodulation system.

**Table 1 sensors-21-06609-t001:** Speed test results of MLE algorithm.

MLE Finds the Peak Interval (nm)	MLE Finds Peak Range (μm)	Speed Test Results (μs)
10	600	2.01350 × 10^3^
	100	317.45
	10	16.24
	1	1.12
1	600	3.945828 × 10^4^
	100	4.32464 × 10^3^
	10	226.75
	1	21.24
0.1	600	8.6255346 × 10^5^
	100	6.586125 × 10^4^
	10	5.476215 × 10^3^
	1	274.89

## Data Availability

The data that support the findings of this study are available from the corresponding author upon reasonable request.

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
