# Peer review of "A High-Speed Demodulation Technology of Fiber Optic Extrinsic Fabry-Perot Interferometric Sensor Based on Coarse Spectrum"

_sensors, 2021, doi:10.3390/s21196609_

Round 1
Reviewer 1 Report
The authors introduce a new method about fast real-time demodulation in the paper “A High-Speed Demodulation Technology of Fiber Optic Extrinsic Fabry-Perot Interferometric Sensor Based on Coarse Spectrum”. However, there are so many spelling, grammatical and commonsense mistakes in the paper that it is impossible to read it smoothly. Besides, the innovation claimed in the paper is a common method of handling data. I would reject the paper in its current form. My comments are as follows.
- In the abstract, “roughly estimation” is proposed to be changed to “rough estimation”.
- The sentence “At present, intensity demodulation method [10-13]…” lacks an article “the” before “intensity demodulation method”.
- The authors think that the intensity demodulation method is a common high-speed method in the sentence “At present, intensity demodulation method [10-13] is a common high-speed measurement method for EFPI sensors.” However, the author does not list references for the most recent years. I would suggest that the author list new references.
- The author mentions “the three-wavelength intensity demodulation method proposed by Markus Schmidt has a speed of 80 kHz [14]”, but I did not find 80 kHz in the references. Could the author please explain where did they find the number of 80 kHz?
- The sentence “Because the frequency spectrum is independent of the source intensity” is not right in theory.
- With regard to the introduction, the author needs to present the background and superiority of the proposed method, rather than summarising the new method in one sentence. Also, the introduction should be logical and preferably divided into paragraphs.
- Typo in the title “2.1 Corse spectral sampling points analysis”.
- Formatting error. “where” does not require two blank spaces and capitalization in the sentence “Where R is the reflectivity of both ends of the Fabry-Perot cavity”
- Ir(λ) is rewritten as Equation (2), then Equation (2) should be a relation on Ir(λ). Also, what does Sn denote as?
- What are the ASE and SMF in Figure 1? The first time it appears no abbreviations can be used, also F-P does not appear previously. The meaning of I0 (λ) is also not given.
- The “it” in the sentence “It shows that the demodulation of cavity length is essentially an estimation of the frequency of the cosine signal in Equation (2)” is not clear what does it refers to.
- Well-known formulas need not be listed, such as the Nyquist sampling in Eq. (3).
- Please give details of the procedure of how to obtain f = 2L/C by Eq. (1).
- For N, the author needs to list other parameters to illustrate N > 33.5, rather than just saying N > 33.5.
- The first appearance of MLE in the title “2.2 FFT and MLE fusion algorithm” requires a full name.
- The authors think the contours of the dense spectrum and coarse spectrum are almost the same in Figure 2(b). However, the results in Figure 2b are not “almost the same”.
- For “Therefore, the demodulation result of the Fourier transform algorithm is only a rough estimation of the cavity length”, the authors need to show how an estimate of the cavity length can be obtained from the resolution results in Figure 2.
- The author's description seems contradictory, using MLE to improve accuracy, but then reducing the amount of data to improve speed. How can the authors reduce the amount of data and increase accuracy at the same time?
- The meaning of L' in Eq. (5) needs to be indicated, and also Eq. (5) needs to end with a period.
- The meaning of g in Eq. (6) needs to be indicated. “where” does not require two blank spaces and capitalization in the sentence “Where n is the number of samples”
- There is no basis for the authors’ conclusion “has higher resolution, stronger noise suppressing ability and higher demodulation accuracy under the condition of coarse spectral sampling” based on the spectral peaks; these conclusions are not necessarily linked to the peaks. Also, the real-time demodulation speed cannot be illustrated by Fig. 3.
- For the sentence “With the decrease of spectral resolution and samples, the influence of noise on demodulation is intensified”, the logic is wrong. The effect of noise, compared to high resolution and more samples, should be weakened, not enhanced.
- Eq. (7) is wrong.
- As for the experimental results in Figure 12, the available experimental results do not demonstrate an increase in accuracy and resolution. I suggest that the authors add experiments comparing the length of each algorithm with the real cavity.
- The author needs to explain the detail how do they obtain the data of 15 nm and 0.005%.
Author Response
Dear reviewer:
We would like to thank you for allowing us to resubmit a revised draft of our manuscript. We appreciate the time and effort that you dedicated to providing feedback on our manuscript and are grateful for the insightful and valuable comments on our paper. According to the comments and concerns provided in your letter, we will respond one by one in the uploaded file (Response to Reviewer 1 comments).
Thank reviewer for the comments, they are really helpful to further improve the quality of our manuscript.
Your sincerely,
Peng Zhang, Ying Wang, Yuru Chen, Xiaohua Lei, Yi Qi, Jianghua Feng, and Xianming Liu

Reviewer 2 Report
This manuscript describes a fiber optic FP interferometer using double step calculation of cavity length.
It was well organized and the theory and experimental results were technically sound.
I would like to recommend this manuscript as accepted in Sensors.
However, the followings should be considered in revised version.
(1) How long does it take to calculate the cavity length using the MLE algorithm?
(2) In case of using channels of DWDM more than 40, What about the measurement performance of the system?
(3) The FFT result of Fig. 12 is the one used by the typical FFT. Is it not improved if the zero-padding technique is used?
Author Response
Dear reviewer:
We would like to thank you for allowing us to resubmit a revised draft of our manuscript. We appreciate the time and effort that you dedicated to providing feedback on our manuscript and are grateful for the insightful and valuable comments on our paper. According to the comments and concerns provided in your letter, we will respond one by one in the uploaded file (Response to Reviewer 2 comments).
Thank reviewer for the comments, they are really helpful to further improve the quality of our manuscript.
Your sincerely,
Peng Zhang, Ying Wang, Yuru Chen, Xiaohua Lei, Yi Qi, Jianghua Feng, and Xianming Liu

Reviewer 3 Report
Title: A High-Speed Demodulation Technology of Fiber Optic Extrinsic Fabry-Perot Interferometric Sensor Based on Coarse Spectrum
The manuscript proposed a fast-process technique to analysis Fabry-Perot interferometer sensors' signal rapidly. Detail schematic, structure layout and operating principle of the setup has been experimentally conducted and reported.
Some kind comments and suggestions for the authors to make amendments to the manuscript:
- Figure 1 to label all the parts such as the circulator, etc.
- Although not drawn to scale, Figure 1 bottom inset, the right arrow Ir(lamdba) to shift up to the core of the SMF.
- Equations in the manuscript to be formatted to fraction and script for readers to read the equation more easily. Examples, line 54 (1), line 73 (4), etc.
- Please check for technical spelling errors throughout the manuscript. Example, line 50 sub-heading "Corse" to "coarse".
- Line 108, "The shortcoming lies in the rapid increase of computation, which affects the real-time demodulation speed of the system."
- Will be useful to provide qualitative measurement for reader to understand the basic computer configuration setup and compare the time spent for DFT and MLE.
- How much percent of time is saved for example?
- Will it be better to include some qualitative in the introduction as well for past techniques, if available.
- Will be useful to provide qualitative measurement for reader to understand the basic computer configuration setup and compare the time spent for DFT and MLE.
- Line 181, 197, are these equations necessary equations?
- To provide high resolution image for all of the graphs / plots. Font type and size of those image should changed to show clearly the results.
- Figure 9:
- What is the FPGA brand/model/configuration used in the experiments? Can this affects the demodulation speed?
- The supporting PC might also be key influence for high speed demodulation if high performance parts are used.
- The red font and dashed lines might be hard to read for some readers.
- What is the FPGA brand/model/configuration used in the experiments? Can this affects the demodulation speed?
Non-technical comments:
- Line 55, 59, 69, 102, etc. defining the parameters should start with lower case "where".
- Line 97, 184, etc. to end with full stop when the sentence stopped.
- Check for consistent:
- Spacing between value and SI unit for readability. Some of the equations without proper spacing are hard to read.
- Subscript.
- Words in the middle with upper case should be changed to lower case, if they are not supposed to be. Example Line 51, lower case "according".
- Spacing between value and SI unit for readability. Some of the equations without proper spacing are hard to read.
Thank you.
Author Response
Dear reviewer:
We would like to thank you for allowing us to resubmit a revised draft of our manuscript. We appreciate the time and effort that you dedicated to providing feedback on our manuscript and are grateful for the insightful and valuable comments on our paper. According to the comments and concerns provided in your letter, we will respond one by one in the uploaded file (Response to Reviewer 3 comments).
Thank reviewer for the comments, they are really helpful to further improve the quality of our manuscript.
Your sincerely,
Peng Zhang, Ying Wang, Yuru Chen, Xiaohua Lei, Yi Qi, Jianghua Feng, and Xianming Liu

Round 2
Reviewer 1 Report
The authors answer all the questions well.